# Urolithin A Inactivation of TLR3/TRIF Signaling to Block the NF-κB/STAT1 Axis Reduces Inflammation and Enhances Antioxidant Defense in Poly(I:C)-Induced RAW264.7 Cells

**DOI:** 10.3390/ijms23094697

**Published:** 2022-04-23

**Authors:** Wen-Chung Huang, Chian-Jiun Liou, Szu-Chuan Shen, Sindy Hu, Jane C-J Chao, Chien-Yu Hsiao, Shu-Ju Wu

**Affiliations:** 1Research Center for Food and Cosmetic Safety, Graduate Institute of Health Industry Technology, College of Human Ecology, Chang Gung University of Science and Technology, Taoyuan City 33303, Taiwan; wchuang@mail.cgust.edu.tw; 2Division of Allergy, Asthma, and Rheumatology, Department of Pediatrics, Chang Gung Memorial Hospital, Linkou, Taoyuan City 33303, Taiwan; ccliu@mail.cgust.edu.tw; 3Department of Pediatrics, New Taipei Municipal Tu Cheng Hospital, Chang Gung Memorial Hospital, and Chang Gung University, New Taipei City 23678, Taiwan; 4Department of Nursing, Division of Basic Medical Sciences, Research Center for Chinese Herbal Medicine, and Graduate Institute of Health Industry Technology, Chang Gung University of Science and Technology, Taoyuan 33303, Taiwan; 5Graduate Program of Nutrition Science, National Taiwan Normal University, 88 Ting-Chow Rd, Sec 4, Taipei 11677, Taiwan; scs@ntnu.edu.tw; 6Department of Cosmetic Science, College of Human Ecology, Chang Gung University of Science and Technology, Guishan Dist., Taoyuan City 33303, Taiwan; sindyhu@hotmail.com; 7Department of Dermatology, Aesthetic Medical Center, Chang Gung Memorial Hospital, Linkou, Taoyuan City 33303, Taiwan; 8School of Nutrition and Health Sciences, College of Nutrition, Taipei Medical University, 250 Wu-Hsing Street, Taipei 11031, Taiwan; chenjui@tmu.edu.tw; 9Department of Nutrition and Health Sciences, Research Center for Chinese Herbal Medicine, College of Human Ecology, Chang Gung University of Science and Technology, Taoyuan City 33303, Taiwan

**Keywords:** urolithin A, poly(I:C), TLR3, NF-κB, MAPK

## Abstract

Urolithin A is an active compound of gut-microbiota-derived metabolites of polyphenol ellagic acid that has anti-aging, antioxidative, and anti-inflammatory effects. However, the effects of urolithin A on polyinosinic acid-polycytidylic acid (poly(I:C))-induced inflammation remain unclear. Poly(I:C) is a double-stranded RNA (dsRNA) similar to a virus and is recognized by Toll-like receptor-3 (TLR3), inducing an inflammatory response in immune cells, such as macrophages. Inflammation is a natural defense process of the innate immune system. Therefore, we used poly(I:C)-induced RAW264.7 cells and attenuated the inflammation induced by urolithin A. First, our data suggested that 1–30 μM urolithin A does not reduce RAW264.7 cell viability, whereas 1 μM urolithin A is sufficient for antioxidation and the decreased production of tumor necrosis factor-α (TNF-α), monocyte chemoattractant protein-1 (MCP-1), and C-C chemokine ligand 5. The inflammation-related proteins cyclooxygenase-2 and inducible nitric oxide synthase were also downregulated by urolithin A. Next, 1 μM urolithin A inhibited the levels of interferon (INF)-α and INF-β. Urolithin A was applied to investigate the blockade of the TLR3 signaling pathway in poly(I:C)-induced RAW264.7 cells. Moreover, the TLR3 signaling pathway, subsequent inflammatory-related pathways, and antioxidation pathways showed changes in nuclear factor-κB (NF-κB) signaling and blocked ERK/mitogen-activated protein kinase (MAPK) signaling. Urolithin A enhanced catalase (CAT) and superoxide dismutase (SOD) activities, but decreased malondialdehyde (MDA) levels in poly(I:C)-induced RAW264.7 cells. Thus, our results suggest that urolithin A inhibits TLR3-activated inflammatory and oxidative-associated pathways in macrophages, and that this inhibition is induced by poly(I:C). Therefore, urolithin A may have antiviral effects and could be used to treat viral-infection-related diseases.

## 1. Introduction

Mammalian Toll-like receptors (TLRs) recognize various pathogen-associated molecular patterns (PAMPs) and play crucial roles in the innate immune system [1]. Polyinosinic-polycytidylic acid (poly(I:C)) is a double-stranded RNA (dsRNA), and the specific ligand of TLR3 recognizes viral dsRNA [2]. Recently, poly(I:C) was shown to be able to activate the TLR3/TRIF pathway in the systemic inflammatory response [3]. The TLR family interacts with PAMPs and recruits a TIR domain-containing adapter inducing interferon (IFN)-β (TRIF), which initiates TLR signaling to activate the transcription factors nuclear factor-κB (NF-κB) and interferon-regulatory factor 3 (IRF3) to induce innate immune responses and inflammatory mediators [4]. The activation of TLRs immediately regulates cytokines, chemokines, and type I IFN production with the subsequent induction of IFN-responsive genes, such as antiviral genes and other inflammatory mediators, to protect the host from microbial infection [5,6]. This process initiates the host’s defensive responses, such as inflammation and antioxidants, to protect the host from microbial infection [1].

Normally, NF-κB is localized in the cytoplasm and binds to the inhibitory κB (IκB) proteins. A variety of inflammatory stimuli induce IκB phosphorylation by kinases, leading to the degradation of IκB. The phosphorylation of IκBα leads to NF-κB activation, and the active NF-κB translocates into the nucleus and activates the target genes. Through TLR3 signaling, poly(I:C) mediates the phosphorylation of IκB and NF-κB activation [7], which then modulates the innate immune and oxidative stress responses. Studies have indicated that the activation of NF-κB increases cellular oxidative stress [8,9]. In addition, studies have indicated that interactions between the nuclear factor erythroid 2-related factor 2 (Nrf2) and NF-κB pathways in cells cause oxidative stress. Thus, NF-κB modulates Nrf2 transcription and activity, and the NF-κB pathway can have both anti- and pro-oxidant roles in the setting of oxidative stress [8,10]. On the other hand, Kelch-like ECH-associating protein 1 (KEAP1) is an IκB kinase (IKK)β E3 ubiquitin ligase that can prevent NF-κB pathway activation. As such, NF-κB can have different roles in Nrf2 expression and activity [11]. The TLR3-induced generation of intracellular reactive oxygen species (ROS) is required for the activation of NF-κB, and the phosphorylation of the IRF3-induced signal transducer and the activator of transcription 1 (STAT1) contributes to the generation of inflammatory mediators and innate immune responses in macrophages. Alternatively, TLR3-ROS signaling plays a role in innate immune responses by activating STAT1 [12]. The ROS molecules induce proteins and lipid peroxidation, as well as DNA damage. NF-κB activation has been shown to increase manganese superoxide dismutase (Mn-SOD) levels in TNF-α-treated Ewing’s sarcoma cells. Thus, the activation of the NF-κB pathway may also induce antioxidant activity [13]. In addition, STAT1 and NF-κB signaling may induce the transcriptional activation of type I IFNs (IFN-α/β) in activated macrophages [14,15], as well as many antiviral genes for survival [16].

In particular, TRIF by poly(I:C), which phosphorylates IRF3 to induce the transcription of pro-inflammatory cytokines and type I IFNs, increases the expression of cytokines, such as IFN-β [17,18]. Studies have indicated that poly(I:C)-induced IRF3 phosphorylation immediately activates mitogen-activated protein kinase (MAPK) signaling. Simultaneously, TRIF-dependent signaling can activate MAPK signaling [19,20,21]. There are three MAPKs in mammalian cells: c-Jun N-terminal kinase (JNK), extracellular signal-regulated kinase (ERK), and p38 MAPK. A variety of transduced signals induce MAPK phosphorylation, activating MAPK in the nucleus to mediate cellular processes, including stress responses and cancer progression. In addition, studies have shown that MAPK mediates important physiological and pathological functions in innate immunity [22,23]. TRIF induces IRF3, NF-κB, and MAPK kinase activation, affecting viral infections [24]. Moreover, poly(I:C) could activate TLR3-induced cyclooxygenase-2 (COX-2) protein expression and increase JNK, ERK, and p38 MAPK phosphorylation to promote an inflammatory process in the brain [25].

Pomegranate (*Punica granatum* L.) contains polyphenols, including punicalagin, punicalin, ellagic acid, and gallic acid, which are rich in the fruit, leaves, and peel [26]. However, the bioavailability of dietary polyphenol is low, and ellagic acid and related polyphenols from the consumption of pomegranates are metabolized to urolithin A by gut microbes [27,28]. Thus, urolithin A is a microflora-derived metabolite. Studies have found that a mitophagy activator improves muscle mitochondrial and cellular health in older humans [29]. By improving mitochondrial functioning and mitigating MAPK/NF-κB/Akt signaling to decrease ROS and pro-inflammatory cytokine levels, urolithin A attenuates metabolic diseases in multiple tissues, preventing Alzheimer’s disease, type 2 diabetes mellitus, non-alcoholic fatty liver disease, and attenuated ox-LDL-induced cholesterol accumulation [30,31]. Interestingly, urolithin A is a potential neuroprotective agent, decreasing oxidative stress and increasing antioxidant enzymes in H_2_O_2_-actived Neuro-2a cells [32]. Through the modulation of the gut microbiota, urolithin A also decreases body weight gain and improves inflammation and dysfunctional lipid metabolism in high-fat-diet-induced obese mice [33].

The antiviral mechanisms of urolithin A remain unclear because its therapeutic effects are not fully explained by its interaction with TLR3. Previous studies have shown antioxidant, anti-inflammation, and anti-aging effects, but the direct action of urolithin A inhibition by poly(I:C) and its attenuated potential on innate immunity remains unexplored. The present study aimed to explore the effects of urolithin A on the poly(I:C)-induced activation of TLR3’s antiviral effects, counterbalancing NF-κB/MAPK’s inflammatory responses and enhancing the NRF2 antioxidation pathway. We found that urolithin A inhibits poly(I:C)-induced IFN-α and IFN-β levels and induces phosphorylated STAT1 (pSTAT1) downstream of these cytokines. In addition, urolithin A, accompanied by the ERK/MAPK inhibitor (PD98059), reduces the TNF-α, MCP-1, and INF-β levels induced by poly(I:C) in RAW264.7 macrophages. Therefore, the decrease in poly(I:C)-induced IFN-α and IFN-β levels after urolithin A treatment could depend on the inhibited phosphorylation of IRF3.

## 2. Results

### 2.1. Urolithin A Inhibited Inflammatory Cytokines and Inactived TLR3 Pathway Protein Expression in Poly(I:C)-Stimulated RAW264.7 Cells

Urolithin A at concentrations of ≥30 μM showed no significant cytotoxicity in RAW264.7 cells, but concentrations of ≥60 μM significantly reduced cell numbers (Figure 1B). Therefore, all experiments used 1–30 μM urolithin A. When RAW264.7 cells were treated with urolithin A in six-well plates and stimulated with poly(I:C) (1 μg/mL) at concentrations of ≥1 μM, this significantly reduced inflammatory cytokines, including TNF-α, MCP-1, and CCL-5, compared to poly(I:C) alone (*p* < 0.01; Figure 1C–E). Interestingly, ≥1 μM urolithin A significantly inhibited TLR3 protein expression (*p* < 0.05), and ≥10 μM urolithin A significantly decreased TRIF and pIRF protein expression compared to poly(I:C) alone (*p* < 0.05; Figure 2A–F). A previous study found that a ligand of TLR3 contributes to the viral-infection-induction of chemokine CCL5/RANTES secretion by airway epithelial cells to attract inflammatory cells [34]. In addition, cytokines TNF-α and MCP-1, produced by macrophages and dendritic cells, are important for innate immunity against viruses. TRIF and IRF3 are activated in poly(I:C)-exposed macrophages, leading to the production of inflammatory mediators [34,35]. Our data showed that TLR3, TRIF, and pIRF3 are activated in poly(I:C)-exposed macrophages, leading to the production of TNF-α, MCP-1, and CCL-5. Urolithin A significantly inhibited TLR3, TRIF, and pIRF3 protein expression in poly(I:C)-stimulated RAW264.7 cells compared to poly(I:C) alone.

### 2.2. Urolithin A Blocked the NF-κB/STAT1 Pathway to Inhibit the Expression of Inflammatory Mediators in Poly(I:C)-Induced RAW264.7 Cells

Urolithin A concentrations of ≥3 μM significantly blocked NF-κB activation (Figure 3A–C) and concentrations of ≥10 μM inhibited pIκB expression (Figure 3D,E) compared to poly(I:C) alone. Furthermore, urolithin A concentrations of ≥3 μM significantly suppressed STAT1 phosphorylation compared to poly(I:C) alone (*p* < 0.01). Previous studies have indicated that poly(I:C)-induced NF-κB and STAT1 activation promotes IFN-β secretion in aortic valve interstitial cells [35]. In addition, poly(I:C)-induced TLR3 expression and the activation of the STAT1 pathway regulates the innate immune response, resulting in the production of IFN-α/β in epidermal keratinocytes [36]. Therefore, we evaluated the ability of urolithin A to inhibit NF-κB and STAT1 in relation to the suppression of IFN-α/β secretion and decreased inflammatory mediators in poly(I:C)-stimulated RAW264.7 cells. Immunofluorescent staining to investigate urolithin A showed that it could inhibit the translocation of NF-κB p65 from the cytoplasm into the nucleus when cells were pre-cultured with or without urolithin A (1, 3, 10, 30 μM) for 1 h and added poly(I:C) (1 μg/mL) for 30 min. Urolithin A increased the p65 subunit retention in the cytoplasm in poly(I:C)-stimulated RAW264.7 cells (Figure 3A). Moreover, we used immune blots to analyze NF-κB and IκB protein expression and found that urolithin A decreased NF-κB and IκB phosphorylation compared to poly(I:C) alone (Figure 3B–E). Importantly, urolithin A at concentrations of ≥1 μM strikingly decreased STAT1 phosphorylation compared to poly(I:C) alone (Figure 4A,B). We also found that urolithin A at ≥10 μM decreased the expression of the inflammatory mediators COX-2 and iNOS compared to poly(I:C) alone (Figure 4C–F). Our results indicate that urolithin A inhibits inflammatory mediator expression in poly(I:C)-induced RAW264.7 cells via the blockade of the NF-κB and STAT1 pathways.

### 2.3. Urolithin A Suppressed the ERK/MAPK Pathway and the ERK Inhibitor Decreased Cytokine Secretion in Poly(I:C)-Stimulated RAW264.7 Cells

Next, we evaluated whether urolithin A inhibits MAPK activation and MAPK inhibitors decrease inflammation in poly(I:C)-induced RAW264.7 cells. First, to evaluate MAPK signaling protein expression, cells were pre-cultured with or without urolithin A (1, 3, 10, 30 μM) for 1 h and added poly(I:C) (1 μg/mL) for 30 min or 24 h. The collected protein samples were analyzed by immunoblot. Urolithin A concentrations of ≥3 μM significantly decreased pERK1/2 expression compared to poly(I:C) alone (*p* < 0.05; Figure 5A,B). Nevertheless, urolithin A did not affect pp38 or pJNK protein expression (Figure 5C–F). Next, we evaluated the effect of the ERK/MAPK inhibitor on urolithin A-induced changes to inflammatory cytokine concentrations in poly(I:C)-induced RAW264.7 cells. RAW264.7 cells were pre-cultured with urolithin A (3 and 10 μM) and/or 10 μM ERK1/2 inhibitor PD98059 before incubation with poly(I:C) (1 μg/mL). After 24 h, the collected supernatant was evaluated by ELISA for cytokine concentrations. Although urolithin A at 10 μM + poly(I:C) significantly decreased MCP-1 and IFN-β (*p* < 0.01 and *p* < 0.05, respectively), that urolithin A at 10 μM decreased TNF-α utility is equivalent to the ERK inhibitor + poly(I:C) alone. Urolithin A at 3 μM + poly(I:C) cannot achieve the same effect as the ERK1/2 inhibitor + poly(I:C) alone. In addition, the ERK1/2 inhibitor + poly(I:C) + urolithin A at concentrations of ≥3 μM markedly decreased the concentrations of TNF-α and MCP-1 levels, and urolithin A ≥10 μM significantly decreased IFN-β levels compared to the poly(I:C) + ERK1/2 inhibitor (*p* < 0.01). Moreover, the ERK1/2 inhibitor + poly(I:C) + urolithin A at concentrations of ≥3 μM significantly decreased TNF-α, MCP-1, and IFN-β levels compared to poly(I:C) + urolithin A alone (*p* < 0.01). These results showed there is an additive effect with a combination of urolithin A and the ERK1/2 inhibitor. Furthermore, Uro-A at ≥1 μM remarkably inhibited IFN-α and IFN-β concentrations compared to poly(I:C) alone in poly(I:C)-induced RAW264.7 cells (*p* < 0.01) (Figure 6D,E). Therefore, we suggest that, in poly(I:C)-stimulated RAW264.7 cells, urolithin A attenuated the NF-κB/STAT1 and ERK/MAPK pathways to inhibit the expression of inflammatory mediators (COX-2 and iNOS) and decrease inflammatory cytokine (TNF-α and MCP-1) and IFN-α/β concentrations.

### 2.4. Urolithin A Elevated Nrf2 Transcriptional Regulation and Enhanced Antioxidant Cytoprotective Defense in Poly(I:C)-Stimulated RAW264.7 Cells

Immunofluorescent staining was performed to evaluate urolithin A’s regulation of the antioxidant transcription factor Nrf2. RAW264.7 cells were pre-cultured with urolithin A (1–30 μM) for 1 h and then with added poly(I:C) (1 μg/mL) for 30 min. Nrf2 was retained in the cytoplasm of poly(I:C)-stimulated RAW264.7 cells, whereas urolithin A at concentrations of ≥3 μM affected Nrf2 translocation from the cytoplasm into the nucleus (Figure 7A). Furthermore, urolithin A at concentrations of ≥1 μM significantly enhanced the CAT and Mn-SOD activities compared to poly(I:C) alone (Figure 7B,C). MDA was also significantly reduced by urolithin A at concentrations of ≥1 μM compared to poly(I:C) alone (Figure 7D). According to the above results, we suggest that urolithin A has a Nrf2-mediated dose response to enhanced antioxidant defense in poly(I:C)-stimulated RAW264.7 cells.

## 3. Discussion

Human immune cells, including macrophages, elicit IFN-α/β production to activate natural killer (NK) cells via TLR3 receptors on the endosomal membrane, which are sensors of viral dsRNA and poly(I:C) [37]. In viral infection, TLR3 recognizes dsRNA and, through TRIF, transmits signals to activate the transcription factor IRF3. The phosphorylation of IRF3 results in its translocation into the nucleus, leading to IFN-α/β and inflammatory cytokine production and antiviral immune responses [38]. Therefore, it is worth exploring the TLR3-mediated type I IFN signaling pathway. Here, we demonstrated that urolithin A can prevent TLR3-mediated IFN-α/β antiviral responses (Figure 2 and Figure 6D,E).

Previous studies have shown that poly(I:C), through the activation of the NF-κB pathway, induces iNOS and COX-2 overexpression, subsequently producing proinflammatory cytokines [39,40]. Urolithin A is an excellent inhibitor, inactivating NF-κB (Figure 3) and attenuating the expression of proinflammatory mediators (e.g., iNOS, COX-2) (Figure 4C,D) to achieve decreased cytokine (TNF-α, MCP-1, and CCL-5) secretion (Figure 1C–E). Furthermore, NF-κB has crosstalk with the STAT and IRF3 pathways in the antiviral response. Studies have indicated that NF-κB and IRF3 translocate into the nucleus in poly(I:C)-induced MEF cells [41]. Poly(I:C)-activated NF-κB simultaneously induces IRF3 and STAT1, leading to INF-α/β secretion, and IKKα is activated by dsRNA-mediated STAT1 phosphorylation in antiviral signaling by the innate immune system [7]. In the inflammatory response, transcription factors NF-κB and STAT1 are important regulators of inflammatory cytokine production and inflammatory cell infiltration. During the infection, STAT1 is a TLR-mediated antibody response and B cell differentiation [42]. STAT1 plays an essential role in the TLR-mediated antibody response of the marginal zone during inflammation and infection [43]. Thus, STAT1 is an important target in preventing or treating viral-infection-induced inflammation. In the present study, we demonstrated that poly(I:C)-induced NF-κB and STAT1 phosphorylation, and pre-treatment with urolithin A efficiently inhibited pNF-κB and pSTAT1 protein expression (Figure 3 and Figure 4A). We suggest that urolithin A’s blockade of NF-κB and STAT pathways is mediated by TLR3 in poly(I:C)-induced RAW264.7 cells.

Macrophages play important roles in immunity and various inflammatory diseases via the production of cytokines and the inflammatory mediators iNOS and COX-2. Many studies have elucidated the relationship between cytokines and inflammatory mediators involved in the MAPK inflammatory pathway [40]. Previous studies have indicated that poly(I:C) binds to endosomes in the cells and induces transcription factor activation, resulting in the secretion of antiviral IFN-I, which feeds back to antiviral gene replication. Furthermore, MAPK/p38 and ERK are required for poly(I:C)/TLR3-mediated cytokine production [44,45]. This result indicates that poly(I:C) increases iNOS and COX-2 overexpression and induces cytokine production. In the present study, we investigated the mechanisms underlying poly(I:C)-induced inflammatory and antiviral MAPK signaling. First, urolithin A significantly attenuated pERK/MAPK expression, but we did not observe the same expression in pP38 and pJNK/MAPK (Figure 5). In addition, when we blocked the ERK/MAPK pathway, we observed anti-inflammatory and antiviral effects. As stated above, reducing ERK/MAPK protein expression may involve the expression of the downstream inflammatory mediator iNOS and COX-2. Urolithin A and the ERK inhibitor perfectly decreased cytokines and IFN-I concentration (Figure 6). These findings demonstrate that urolithin A has antiviral immunity and anti-inflammation effects via the modulation of the ERK/MAPK pathway.

In addition, poly(I:C) has been indicated as activating TLR3, subsequently causing inflammation and IFN-β expression and inducing ROS production [46]. Mounting evidence suggests that transcription factor Nrf2 regulates cellular-defense oxidative stress and prevents inflammation-related diseases. On the other hand, studies have shown that inflammatory responses are the second line in innate immunity, and the activation of TLR3 signaling and Nrf2 expression work together to regulate the innate immune system [47]. The interplay between TLR signaling and the Nrf2 pathway may be beneficial for the prevention or treatment of inflammation. Nrf2 signaling is a major modulator of antioxidant enzymes and attenuates inflammation. Nrf2 signaling can be activated by TLR agonists and enhance antioxidant defense [48]. Previous studies have implicated TLR ligands as inflammation mediators and immuno-modulators through the activation of Nrf2 to regulate downstream antioxidant enzymes [49]. Importantly, activated Nrf2 limits IκB phosphorylation and keeps NF-κB in the cytoplasm, reducing inflammatory mediators, such as iNOS, COX-2, and cytokines [50]. However, whether the interaction between Nrf2 and TLR signaling regulates the antioxidant and anti-inflammation activity of macrophages infected by poly(I:C) remains unknown. The oxidative-stress-related enzymes SOD and CAT and the lipid peroxidation product MDA were detected in this study. Furthermore, urolithin A promoted Nrf2 translocation from the cytoplasm into the nucleus, enhancing SOD and CAT activities to reduce MDA levels in poly(I:C)-induced macrophages (Figure 7). We also observed that NF-κB, IκB phosphorylation, and inflammatory mediators were inhibited by urolithin A. These results suggest that urolithin A depends on the activation of Nrf2 to enhance antioxidative defense and anti-inflammatory actions.

Taken together, the data indicate that urolithin A can inhibit TLR3 expression and reduce IFN-α/β secretion by suppressing IRF3 phosphorylation. Moreover, urolithin A suppressed the activation of NF-κB/STAT1 and ERK/MAPK signaling, reducing inflammatory mediators (iNOS, TNF-α, MCP-1, and CCL-5) in poly(I:C)-stimulated macrophages. Importantly, urolithin A significantly modulated Nrf2 to increase antioxidant enzymes (SOD and CAT) and decrease the inflammatory responses in poly(I:C)-stimulated macrophages (Figure 8).

## 4. Materials and Methods

### 4.1. Materials

Urolithin A (≥97% purity by HPLC, Figure 1A) was purchased from Sigma-Aldrich (St. Louis, MO, USA), cell viability assays (MTT), and poly(I:C) were purchased from InvivoGen (San Diego, CA, USA). Enzyme-linked immunosorbent assay (ELISA) kits were purchased from R&D Systems (Minneapolis, MN, USA). DAPI solution was purchased from Sigma-Aldrich (St. Louis, MO, USA). The inhibitor PD98059 was purchased from Enzo Life Sciences (Farmingdale, NY, USA). Antibodies against β-actin, TLR3, TRIF, IRF, STAT1, NF-κB, I-κB, COX-2, iNOS, Nrf2, and phosphorylated IRF (pIRF), STAT1, NF-κB (pNF-κB), and IκB (pIκB) were purchased from Santa Cruz Biotechnology (Santa Cruz, CA, USA). Antibodies against JNK, ERK, p38, and phosphorylated JNK (pJNK), ERK (pERK), and p38 (pp38) were purchased from Millipore (Billerica, MA, USA).

### 4.2. Preparation of Urolithin A and Cell Culture

The urolithin A stock solution was prepared at 100 mM and stored at −20 °C. DMSO was added to the medium at a final concentration of ≤0.1% as described previously [33]. The mouse macrophage RAW264.7 cell line was purchased from Bioresource Collection and Research Center (BCRC, Taiwan). Cells were cultured in Dulbecco’s Modified Eagle Medium (Invitrogen-Gibco, Paisley, Scotland) with 10% heat-inactivated fetal bovine serum (FBS; Invitrogen-Gibco, Paisley, Scotland) and 100 U/mL penicillin G, 100 μg/mL streptomycin, and 50 ng/mL gentamycin at 37 °C in a humidified atmosphere of 5% CO_2_. RAW264.7 cells were seeded in serum-free medium for 8 h. Subsequently, RAW264.7 cells were cultured in DMEM medium with 10% FBS. RAW264.7 cells were pre-cultured with or without urolithin A at various concentrations (1, 3, 10, 30 μM) for 1 h, and then with added poly(I:C) (1 μg/mL). After 24 h, the RAW264.7 cells were lysed for Western blot analysis and the supernatant used for ELISA.

### 4.3. Cell Viability Assay

We used MTT kits (Sigma-Aldrich) to assess the inhibitory effect of urolithin A on RAW264.7 cell viability. Cells were seeded in 96-well plates at a concentration of 10^5^ cells/well and urolithin A was added at a concentration of between 1–30 μM for 24 h before culturing at 37 °C for 2 h. After 2 h, MTT solution was added to the plate and a microplate reader (Multiskan FC; Thermo, Waltham, MA, USA) was used at 450 nm to assay cell viability. Each concentration was repeated three times in independent measurements and the inhibitory effect on RAW264.7 cells was reported as a percentage relative to the cells without urolithin A treatment.

### 4.4. ELISA Assay

RAW264.7 cells (10^5^ cells/mL) were pre-cultured with or without urolithin A (1–30 μM) in 24-well plates for 1 h, and then poly(I:C) (1 μg/mL) was added before continuing the culture for 24 h. The levels of TNF-α, MCP-1, CCL-5, INF-α, and INF-β were measured in the supernatant using specific ELISA kits following the manufacturers’ instructions. A microplate reader (Multiskan FC; Thermo) was used at 450 nm to determine the optical density (OD) value.

### 4.5. Preparation of Total Proteins

RAW264.7 cells (2 × 10^5^ cells/mL) were pre-cultured with or without urolithin A (1–30 μM) and stimulated with or without poly(I:C) (1 μg/mL) for 24 h or 30 min in six-well plates to evaluate the total protein and phosphorylated protein content, respectively. We used 300 mL of lysis buffer (Tris–HCl (pH 7, 450 mM); EDTA (1 mM); NaCl (150 mM); DTT (1 mM); NP40 (0.5%); and sodium dodecyl sulfate (SDS, 0.1%)) containing a protease inhibitor cocktail and phosphatase inhibitors (Sigma, St. Louis, MO, USA) to harvest cells. The BCA assay kit (Pierce) was used to quantitate all protein concentrations.

### 4.6. Western Blot Analysis

Proteins were separated on 10% SDS polyacrylamide gels and transferred to polyvinylidene fluoride (PVDF) membranes (Millipore, Billerica, MA, USA). The membranes were incubated overnight at 4 °C with a primary antibody against actin (Sigma) and the protein of interest. The membranes were then washed three times with Tris-buffered saline with Tween 20 (TBST) and incubated at room temperature for 1 h with a secondary antibody. Finally, the proteins were detected using Luminol/Enhancer solution (Millipore) and the base quantitated by the Bio Spectrum 600 system (UVP, Upland, CA, USA).

### 4.7. Immunofluorescence

RAW264.7 cells (2 × 10^5^ cells/mL) were seeded in six-well plates until they reached 50–60% confluence and pre-cultured with or without urolithin A (1–30 μM) for 1 h. Poly(I:C) (1 μg/mL) was added for 30 min before suctioning out the medium. The cells were washed with PBS and fixed with paraformaldehyde (4%, *w*/*v*). The fixed cells were incubated with anti-NF-kB (1:100; Cell Signaling Technology, MA, USA) and NrF2 (1:100; Santa Cruz, CA, USA) antibody overnight at 4 °C. The next day, the medium was removed, and the cells washed with PBS before incubating with secondary antibodies at room temperature for 1 h. The cells were washed with PBS again to remove the dye, and DAPI solution (4′,6-diamidino-2-phenylindole, Sigma) was added to stain the nucleus. A fluorescence microscope (Olympus, Tokyo, Japan) was used to acquire the images. The control groups were treated with poly(I:C) alone, and all experiments were repeated three times.

### 4.8. Antioxidant Defense

Lipid peroxidation produced malondialdehyde (MDA), and the antioxidant enzyme superoxide dismutase (SOD) and catalase (CAT) activities were analyzed as an indicator of the antioxidant defense. First, RAW264.7 cells were treated with urolithin A (1–30 μM) for 1 h, and then stimulated with poly(I:C) (1 μg/mL) for 24 h. Using commercial kits, we analyzed the MDA, SOD, and CAT according to the manufacturer’s instructions (Sigma-Aldrich). A spectrophotometer was used to measure the absorbance at 532 nm for MDA, 450 nm for SOD, and 585 nm for CAT.

### 4.9. Statistical Analysis

Image Lab software (Bio-Rad) was used to quantify the intensity of Western blot bands. Data are presented as the mean ± standard deviation (SD) of at least three independent experiments. We used one-way analysis of variance (ANOVA) followed by Tukey’s post-hoc test. A *p*-value of < 0.05 was considered significant.

## 5. Conclusions

This study first demonstrated that urolithin A may be a TLR3 inhibitor, blocking the NF-κB/STAT1 axis and modulating the Nrf2/NF-κB pathway to enhance antioxidant defense and attenuate inflammatory responses in poly(I:C)-stimulated macrophages.

## Figures and Tables

**Figure 1 ijms-23-04697-f001:**
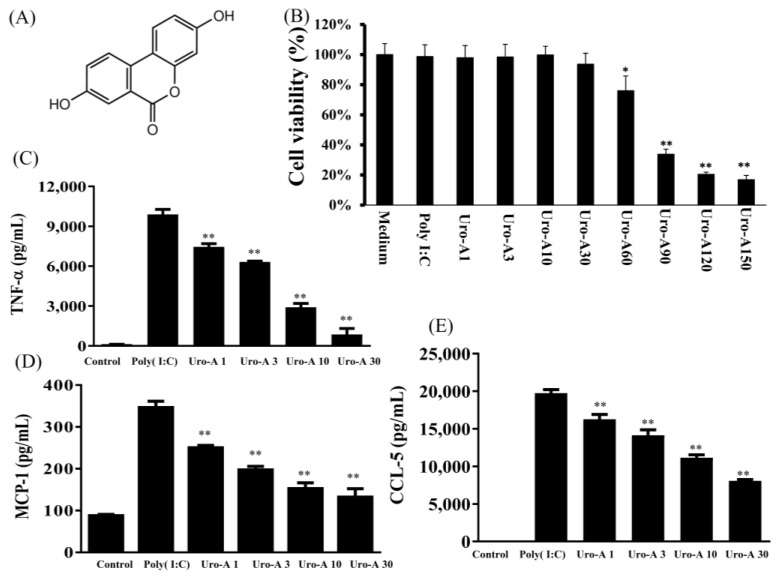
Decreased inflammation-related cytokine secretion. (**A**) Structure of Uro-A. (**B**) Cell viability of MTT. (**C**) TNF-α, (**D**) MCP-1, and (**E**) CCL-5 concentrations. Cells were pre-treated with Uro-A for 1 h, and then with added poly(I:C) (1 μg/mL) for 24 h. Uro-A significantly decreased the inflammatory cytokine concentration in poly(I:C)-stimulated RAW264.7 cells. Data are presented as mean ± SD. * *p* < 0.05, ** *p* < 0.01 compared to RAW264.7 cells stimulated with poly(I:C) alone.

**Figure 2 ijms-23-04697-f002:**
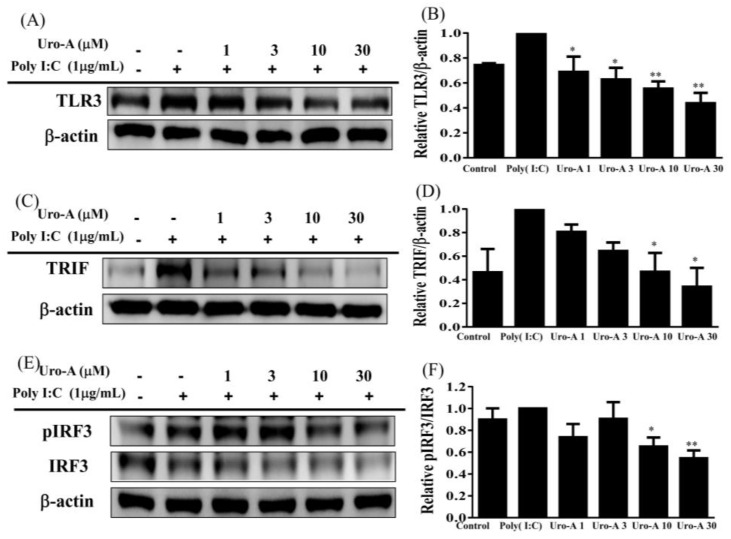
Urolithin A (Uro-A) suppressed TLR3 pathway protein expression in poly(I:C)-stimulated RAW264.7 cells. (**A**) TLR3 protein expression. (**B**) Fold-change in TLR3 expression relative to β-actin expression. (**C**) TRIF protein expression. (**D**) Fold--change in TRIF expression relative to β-actin expression. (**E**) pIRF3 and IRF3 protein expression. (**F**) Fold-change in pIRF3 expression relative to IRF3 expression. RAW264.7 cells were precultured with various concentrations of Uro-A, and then incubated with poly(I:C) (1 μg/mL) for 30 min or 24 h. Data are presented as mean ± SD. * *p* < 0.05, ** *p* < 0.01 compared to RAW264.7 cells stimulated with poly(I:C) alone.

**Figure 3 ijms-23-04697-f003:**
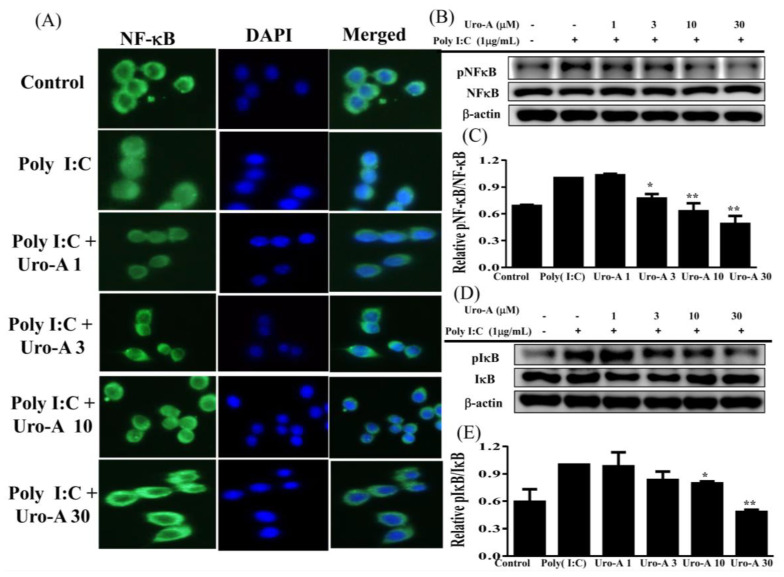
Urolithin A (Uro-A) blocked the NF-κB activation in poly(I:C)-induced RAW264.7 cells. (**A**) DAPI staining showed that Uro-A inhibited NF-κB p65 from translocating from the cytoplasm into the nucleus. Cells were pre-cultured with Uro-A, and then incubated with poly(I:C) (1 μg/mL) for 30 min. Immunofluorescent staining was performed to evaluate NF-κB p65 (green) translocation. Blue is nuclear staining with DAPI. (**B**) pNF-κB and NF-κB protein expression. (**C**) Fold-change in μpNF-κB expression relative to NF-κB expression. (**D**) pI-κB and I-κB protein expression. (**E**) Fold-change in pI-κB expression relative to I-κB expression. RAW264.7 cells were precultured with various concentrations of Uro-A, and then incubated with poly(I:C) (1 μg/mL) for 30 min or 24 h. Data are presented as mean ± SD. * *p* < 0.05, ** *p* < 0.01 compared to RAW264.7 cells stimulated with poly(I:C) alone.

**Figure 4 ijms-23-04697-f004:**
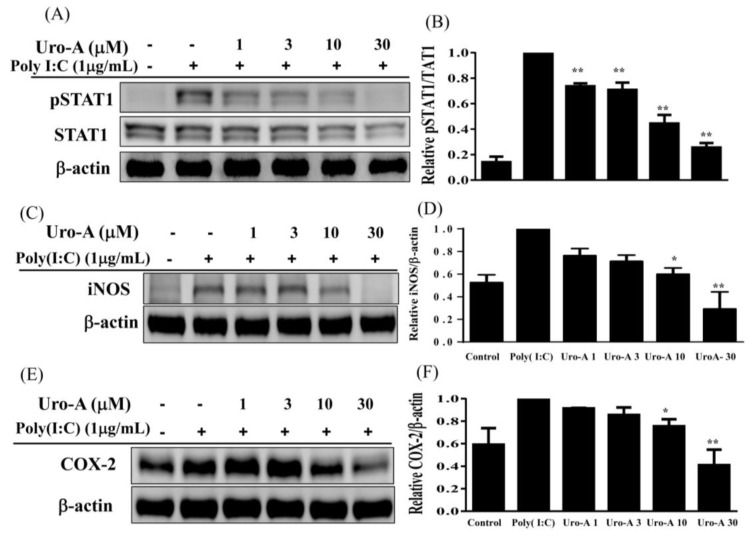
Urolithin A (Uro-A) inhibited the expression of inflammatory mediators in poly(I:C)- induced RAW264.7 cells. (**A**) pSTAT1 and STAT1 protein expression. (**B**) Fold-change in pSTAT1 expression relative to STAT1 expression. (**C**) iNOS protein expression. (**D**) Fold-change in iNOS expression relative to β-actin expression. (**E**) COX-2 protein expression. (**F**) Fold-change in COX-2 expression relative to β-actin expression. RAW264.7 cells were precultured with various concentrations of Uro-A, and then incubated with poly(I:C) (1 μg/mL) for 30 min or 24 h. Data are presented as mean ± SD. * *p* < 0.05, ** *p* < 0.01 compared to RAW264.7 cells stimulated with poly(I:C) alone.

**Figure 5 ijms-23-04697-f005:**
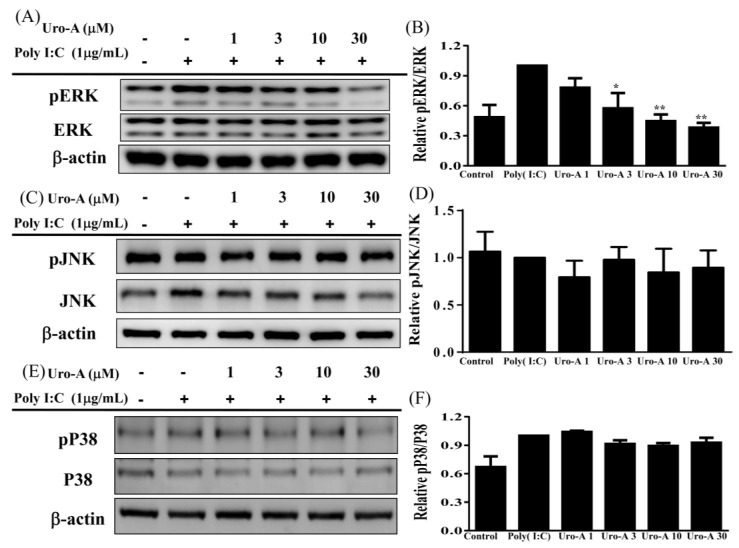
Urolithin A (Uro-A) inhibited inflammatory pERK signaling in poly(I:C)-induced RAW264.7 cells. (**A**) pERK and ERK protein expression. (**B**) Fold-change in pERK expression relative to ERK expression. (**C**) pJNK and JNK protein expression. (**D**) Fold-change in pJNK expression relative to JNK expression. (**E**) pP38 and P38 protein expression. (**F**) Fold-change in pP38 expression relative to P38 expression. RAW264.7 cells were precultured with various concentrations of Uro-A, and then incubated with poly(I:C) (1 μg/mL) for 30 min or 24 h. Data are presented as mean ± SD. * *p* < 0.05, ** *p* < 0.01 compared to RAW264.7 cells stimulated with poly(I:C) alone.

**Figure 6 ijms-23-04697-f006:**
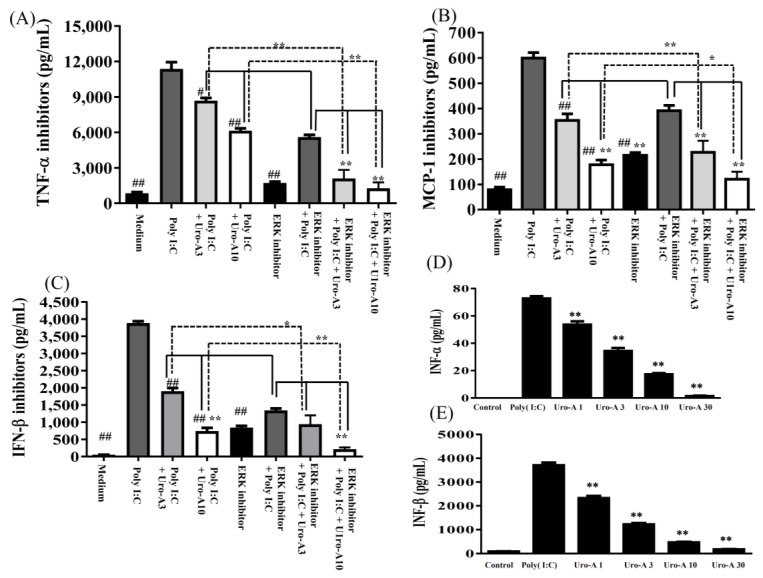
Urolithin A (Uro-A) and the ERK inhibitor suppressed inflammation-related cytokine secretion in poly(I:C)-induced RAW264.7 cells. RAW264.7 cells were pre-cultured with various concentrations of Uro-A, and then incubated with poly(I:C) (1 μg/mL) for 24 h. (**A**) ELISA showed inflammation-related cytokine concentrations of TNF-α, (**B**) MCP-1, and (**C**) IFN-β. (**D**) Uro-A inhibited IFN-α and (**E**) IFN-β concentrations. Data are presented as mean ± SD. ^#^ *p* < 0.05, ^##^ *p* < 0.01 compared to poly(I:C) alone in RAW264.7 cells stimulated with poly(I:C). * *p* < 0.05, ** *p* < 0.01 compared to RAW264.7 cells stimulated with the ERK inhibitor + poly(I:C) alone and the ERK inhibitor + poly(I:C) + Uro-A, respectively.

**Figure 7 ijms-23-04697-f007:**
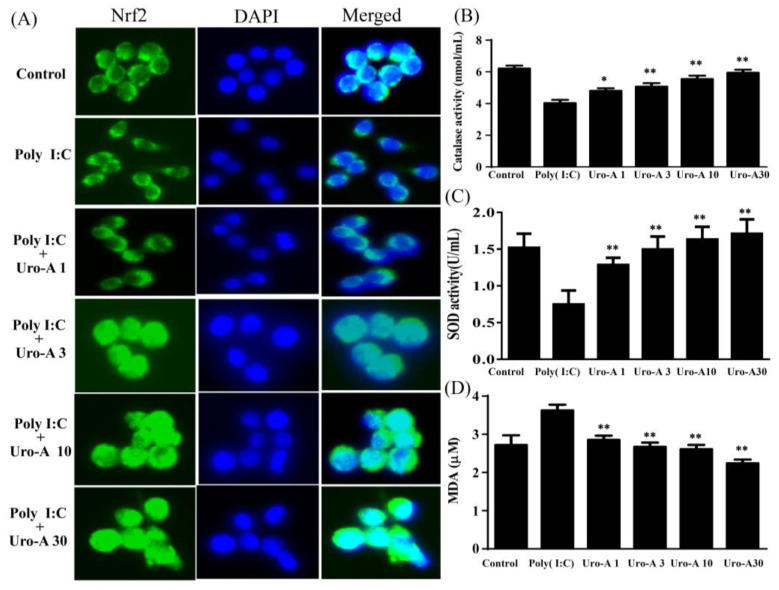
Urolithin A (Uro-A) activated Nrf2 expression and enhanced antioxidant defense in poly(I:C)-stimulated RAW264.7 cells. (**A**) Uro-A promoted Nrf2 (green) translocation from the cytoplasm into the nucleus (DAPI, blue). (**B**) Uro-A increased the activities of antioxidant enzymes SOD and (**C**) CAT, and (**D**) attenuated MDA production. RAW264.7 cells were pre-cultured with different concentrations of Uro-A for 1 h, and then incubated with poly(I:C) (1 ng/mL) for 24 h. Data are presented as mean ± SD. * *p* < 0.05, ** *p* < 0.01 compared to RAW264.7 cells stimulated with poly(I:C) alone.

**Figure 8 ijms-23-04697-f008:**
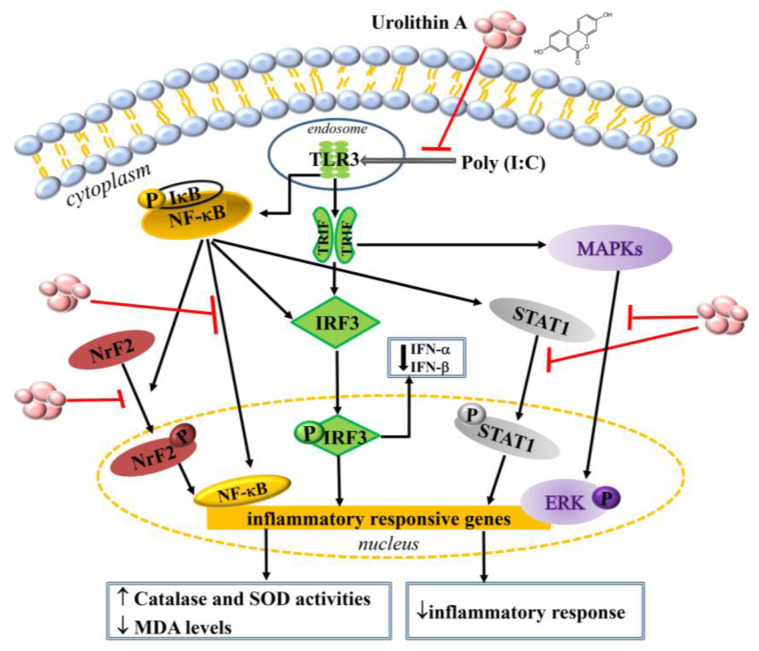
Model explaining the mechanisms of urolithin A’s attenuation of viral-induced inflammation and oxidative stress in macrophages stimulated by poly(I:C).

## Data Availability

The data that support the findings of this study are available from the corresponding author upon reasonable request.

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
