# Peer review of "Urolithin A Inactivation of TLR3/TRIF Signaling to Block the NF-κB/STAT1 Axis Reduces Inflammation and Enhances Antioxidant Defense in Poly(I:C)-Induced RAW264.7 Cells"

_ijms, 2022, doi:10.3390/ijms23094697_

Round 1

Reviewer 1 Report

The aim of this article is to investigate the role of urolithin A (UroA) on Poly (I:C)-induced inflammation and oxidative stress.  It has been demonstrated urolithin A exerts protection against oxidative stress and pro-inflammatory stimuli.  In addition, studies have found treatment with UroA attenuates pro-inflammatory mediator production, at least in part, through suppressing NOX-derived reactive oxygen species-mediated PI3-K/Akt/NF-κB and JNK/AP-1 signaling pathways in LPS-stimulated macrophages.   However, the potential involvement of UroA in regulating macrophage responses to Poly I:C  has not been studied.  Although the experiments are in general well designed, there are several concerns in the presentation and the interpretation of the results.   Following are some specific concerns about the manuscript:

Major concerns:

  1. Figure 3 A. The translocation of the p65 subunit from the cytoplasm to the nucleus by Poly I:C is not clear and consequently figures with Poly I:C and UroA cannot be compared.
  2. Figure 6. Comparison between treatments with UroA+ Poly I:C alone and ERK inhibitor + Poly I:C alone should be indicated.   This figure shows there is an additive effect with combination of UroA and ERK inhibitor.  Therefore, the affirmation that Uro-A suppressed NF-κB/STAT1 and ERK/MAPK pathways is not accurate (Line 230).  UroA attenuates the pathway but does not suppress it.
  3. In Methods. Does the raw cells were kept in reduced serum (for how long?) before stimulated with Poly I:C? Detailed method for IF staining should be included.  Were the cells permeabilized?

Minor concerns

Extensive editing is necessary.  Many sentences are incomplete (line 29, 36…).

Author Response

Manuscript ID: ijms-1674110

Title: Urolithin A Inactive TLR3/TRIF Signals to Block NF-B/STAT 1 Axis Reduce Inflammation and Enhance Antioxidant Defense in Poly (I:C)-Induced RAW264.7 Cells

Authors: Wen-Chung Huang, Chian-Jiun Liou, Shen Szu-Chuan, Sindy Hu, Chao C-J

Jane, Chien‑Yu Hsiao *, Shu-Ju Wu *

Journal: International Journal of Molecular Sciences

Reviewer 1

The aim of this article is to investigate the role of urolithin A (UroA) on Poly (I:C)-induced inflammation and oxidative stress.  It has been demonstrated urolithin A exerts protection against oxidative stress and pro-inflammatory stimuli.  In addition, studies have found treatment with UroA attenuates pro-inflammatory mediator production, at least in part, through suppressing NOX-derived reactive oxygen species-mediated PI3-K/Akt/NF-κB and JNK/AP-1 signaling pathways in LPS-stimulated macrophages.  However, the potential involvement of UroA in regulating macrophage responses to Poly I:C has not been studied. Although the experiments are in general well designed, there are several concerns in the presentation and the interpretation of the results.   Following are some specific concerns about the manuscript: 

Major concerns:

  1. Figure 3 A. The translocation of the p65 subunit from the cytoplasm to the nucleus by Poly I:C is not clear and consequently figures with Poly I:C and UroA cannot be compared.

Responses: Thank for reviewer comment. We replaced clear images in Figure 3 A (Poly I:C alone, page 5).

  1. Figure 6. Comparison between treatments with UroA+ Poly I:C alone and ERK inhibitor + Poly I:C alone should be indicated.   This figure shows there is an additive effect with combination of UroA and ERK inhibitor.  Therefore, the affirmation that Uro-A suppressed NF-κB/STAT1 and ERK/MAPK pathways is not accurate (Line 230).  UroA attenuates the pathway but does not suppress it.

Responses: Thank for reviewer comment.

(1). We compared between treatments with UroA+ Poly I:C alone and ERK inhibitor + Poly I:C alone. In addition, we also indicated ERK inhibitor + Poly I:C +UroA compared to UroA+ Poly I:C alone showed an additive effect in Figure 6. We also corrected Uro-A attenuated NF-κB/ STAT1 and ERK/MAPK pathways (line 243, page 7) in 2.3 section.

This is corrected in 2.3 section (line 228-239, page 7):

When urolithin A at 10 mM + Poly I:C significant decreased MCP-1 and IFN-b (P< 0.01 and p< 0.05, respectively) however, urolithin A at 10 mM decreased TNF-a utility is equivalent to ERK inhibitor + Poly I:C alone. Urolithin A at 3 mM + Poly I:C cannot achieve the same effect as ERK1/2 inhibitor + Poly I:C alone. In addition, ERK1/2 inhibitor + poly(I:C) + urolithin A at concentrations ³ 3 mM markedly decreased the concentrations of TNF-a and MCP-1 levels, and urolithin A ³ 10 mM significant decreased IFN-b levels compared to poly(I:C)+ ERK1/2 inhibitor (p < 0.01).Moreover, ERK1/2 inhibitor + Poly I:C + urolithin A at concentrations ³ 3 mM significant decreased TNF-a, MCP-1 and IFN-b levels compared to poly(I:C)+ urolithin A alone (p < 0.01). These results showed there is an additive effect with combination of urolithin A and ERK1/2 inhibitor.

(2). We improved Figure 6 and Figure 6 Legends describe (page 8).

  1. In Methods. Does the raw cells were kept in reduced serum (for how long?) before stimulated with Poly I:C? Detailed method for IF staining should be included.  Were the cells permeabilized?

Responses: Thank for reviewer comment.

(1). The raw cells were kept in serum free for 8h before stimulated with Poly I:C. We added describe in line 385-387 (section 4.2, page 11).

(2). We added illustrate for IF staining in line 426-429 (section 4.8, page 12).

Minor concerns: Extensive editing is necessary.  Many sentences are incomplete (line 29, 36…). Responses: Thank for reviewer comment. We improved the sentences line 29, 36 and text. We had checked the manuscript by San Francisco Edit as the supplement 1. And we have corrected English grammar and language again in the text.

Supplement 1:

Reviewer 2 Report

In my opinion, the submitted manuscript Urolithin A Inactive TLR3/TRIF Signals to Block NF-κB/STAT 2 1 Axis Reduce Inflammation and Enhance Antioxidant Defense 3 in Poly (I:C)-Induced RAW264.7 Cells meets aims and scope of „International Journal of Molecular Sciences” and may be accepted after the revision.

  1. Due to the poor level of English and formatting, I am not able to evaluate the entire article. The first part of the article (especially abstract and introduction, maybe title and results also) requires an extensive editing of English language and style. The article appears to be partially linguistically corrected, or written by authors with different linguistic abilities.
  2. The formatting of the abstract, introduction, and materials and methods needs to be improved. There are many extra „–“ dash symbols (e.g. line 29, 30, 57, 80, 82, 83,87, 108). Some symbols are incorrectly matched in the text (e.g. symbol of β, in β actin – line 391, or symbol of μ in μg/mL – line 36, or symbol of κ in MAPK/NF-κΒ/Akt – line 113). Numbers  „2” in the H2O2 (line 117) and CO2 (line 360) formulas should be written in subscript. The Latin name of the pomegranate should be written in italics (line 107).
  3. Why figure 8 . is at the end of the article? Authors may expand the conclusion section and insert this figure between the text.
  4. Since the Urolithin A compound is crucial for this article, it’s chemical formula could appear earlier in the manuscript – in the introduction part (not in the results).

Author Response

Manuscript ID: ijms-1674110

Title: Urolithin A Inactive TLR3/TRIF Signals to Block NF-B/STAT 1 Axis Reduce Inflammation and Enhance Antioxidant Defense in Poly (I:C)-Induced RAW264.7 Cells

Authors: Wen-Chung Huang, Chian-Jiun Liou, Shen Szu-Chuan, Sindy Hu, Chao C-J

Jane, Chien‑Yu Hsiao *, Shu-Ju Wu *

Journal: International Journal of Molecular Sciences

Reviewer 2:  In my opinion, the submitted manuscript Urolithin A Inactive TLR3/TRIF Signals to Block NF-κB/STAT 2 1 Axis Reduce Inflammation and Enhance Antioxidant Defense 3 in Poly (I:C)-Induced RAW264.7 Cells meets aims and scope of „International Journal of Molecular Sciences” and may be accepted after the revision.

  1. Due to the poor level of English and formatting, I am not able to evaluate the entire article. The first part of the article (especially abstract and introduction, maybe title and results also) requires an extensive editing of English language and style. The article appears to be partially linguistically corrected, or written by authors with different linguistic abilities.

Responses: Thank for reviewer comment. We extensive editing the text and marked, and corrected the first part of the article and partially linguistically (line 28-50, title, introduction and results). We had checked the manuscript by San Francisco Edit as the supplement 1. And we have corrected English grammar and language again in the text.

Supplement 1:

  1. The formatting of the abstract, introduction, and materials and methods needs to be improved. There are many extra „–“ dash symbols (e.g. line 29, 30, 57, 80, 82, 83,87, 108). Some symbols are incorrectly matched in the text (e.g. symbol of β, in β actin – line 391, or symbol of μ in μg/mL – line 36, or symbol of κ in MAPK/NF-κΒ/Akt – line 113). Numbers „2” in the H2O2 (line 117) and CO2 (line 360) formulas should be written in subscript. The Latin name of the pomegranate should be written in italics (line 107).

Responses: Thank for reviewer comment.

(1). We improved the formatting of the abstract, introduction, and materials and methods.

(2). We had deleted „–“ dash symbols (e.g. line 28-120 and text).

(3). We corrected symbols to match in the text (e.g. symbol of β, in β actin – line 391, or symbol of μ in μg/mL – line 36, or symbol of κ in MAPK/NF-κΒ/Akt – line 113).

(4). We corrected the numbers „2” in the H2O2 (line 118) and CO2 (line 385) formulas written in subscript. And the Latin name of the pomegranate should be written in italics (line 106).

  1. Why figure 8. is at the end of the article? Authors may expand the conclusion section and insert this figure between the text.

Responses: Thank for reviewer comment. We change up the figure 8 in the text (line 357-363 in page 10-11). And we improved conclusions section (line 449-451, page 13).

  1. Since the Urolithin A compound is crucial for this article, it’s chemical formula could appear earlier in the manuscript – in the introduction part (not in the results).

      Responses:

Thank for reviewer’s suggestion. We added “ The chemical structure of Urolithin A (≥97%

purity by HPLC) is illustrated in Figure 1A.” in Introduction (line 110-111, page 3).

Round 2

Reviewer 1 Report

This is a revised version of the paper entitled “Urolithin A Inactivation of TLR3/TRIF Signaling to Block the NF‐κB/STAT1 Axis Reduces Inflammation and Enhances Antioxidant Defense in Poly(I:C)‐Induced RAW264.7 Cells. 

Most of the points raised in the previous review have been addressed.  However, I have few concerns:

  1. In Methods. Does the raw cells were kept in reduced serum (for how long?) before stimulated with Poly I:C?

The authors indicate the raw cells were kept in serum free for 8h before stimulated with Poly I:C.  However, in the manuscript is described: RAW264.7 cells were seeded in serum free medium for 8 hr. Subsequently, RAW264.7 cells were cultured in DMEM medium with 10% FBS.  Please clarified.

  1. In methods. Detailed method for IF staining should be included. Were the cells permeabilized?

The authors partially answered this concern.  It is not indicated if the cells were permeabilized suggesting that this was not performed.  Please explain why this step was omitted since if the target protein is intracellular, it is very important to permeabilize the cells. 

Author Response

Manuscript ID: ijms-1674110

Title: Urolithin A Inactive TLR3/TRIF Signals to Block NF-B/STAT 1 Axis Reduce Inflammation and Enhance Antioxidant Defense in Poly (I:C)-Induced RAW264.7 Cells

Authors: Wen-Chung Huang, Chian-Jiun Liou, Shen Szu-Chuan, Sindy Hu, Chao C-J Jane, Chien‑Yu Hsiao *, Shu-Ju Wu *

Journal: International Journal of Molecular Sciences

Reviewer 1 (Round 2)

Comments and Suggestions for Authors.

This is a revised version of the paper entitled “Urolithin A Inactivation of TLR3/TRIF Signaling to Block the NF‐κB/STAT1 Axis Reduces Inflammation and Enhances Antioxidant Defense in Poly(I:C)‐Induced RAW264.7 Cells.

Most of the points raised in the previous review have been addressed.  However, I have few concerns:

1.The authors indicate the raw cells were kept in serum free for 8h before stimulated with Poly I:C.  However, in the manuscript is described: RAW264.7 cells were seeded in serum free medium for 8 hr. Subsequently, RAW264.7 cells were cultured in DMEM medium with 10% FBS.  Please clarified.

Responses: Thank for reviewer comment. We described in line 385-390 (page 11).

「RAW264.7 cells were seeded in serum-free medium for 8 hr. Subsequently, RAW264.7 cells were cultured in DMEM medium with 10% FBS. RAW264.7 cells were pre-cultured with or without urolithin A at various concentrations (1, 3, 10, 30 μM) for 1 h, and then with added poly(I:C) (1 mg/mL). After 24 h, the RAW264.7 cells were lysed for Western blot analysis and the supernatant used for ELISA.」

2.In methods. Detailed method for IF staining should be included. Were the cells

permeabilized?

The authors partially answered this concern.  It is not indicated if the cells were permeabilized suggesting that this was not performed.  Please explain why this step was omitted since if the target protein is intracellular, it is very important to permeabilize the cells.

Responses: Thank for reviewer comment. We described in line 427-429 (page 12).

Permeabilizing the cells through ice methanol allowed antibodies to pass through the cellular membrane and enter the cell. Hence, we describe as 「The fixed cells were incubated with anti-NF-kB (1:100; Cell Signaling Technology, MA, USA) and NrF2 (1:100; Santa Cruz, CA, USA) antibody overnight at 4°C.」